# Dietary Flavonoid Intake and Chronic Sensory Conditions: A Scoping Review

**DOI:** 10.3390/antiox11071214

**Published:** 2022-06-21

**Authors:** Diana Tang, Yvonne Tran, Giriraj S. Shekhawat, Bamini Gopinath

**Affiliations:** 1Macquarie University Hearing, Faculty of Medicine Health and Human Sciences, Macquarie University, Sydney, NSW 2109, Australia; d.tang@mq.edu.au (D.T.); bamini.gopinath@mq.edu.au (B.G.); 2College of Nursing and Health Sciences, Flinders University, Adelaide, SA 5001, Australia; giriraj.shekhawat@flinders.edu.au; 3Ear Institute, University College London, London WC1X 8EE, UK; 4Tinnitus Research Initiative, 93053 Regensburg, Germany

**Keywords:** flavonoid, dietary intake, macular degeneration, diabetic retinopathy, cataract, glaucoma, hearing loss, tinnitus, adult

## Abstract

Dietary flavonoids have antioxidant, anti-inflammatory, and vascular health benefits, which align with the proposed pathophysiology of age-related eye conditions and hearing problems (hearing loss and tinnitus). This scoping review is based on Arksey and O’Malley’s six-stage framework and aims to summarise current evidence on the association between the dietary flavonoid intake and chronic sensory conditions in adults, and to identify the research gaps in this area. Eligible studies were identified by searching MEDLINE, EMBASE PsycINFO via the OVID platform, and Google Scholar, as well as manually searching the reference lists of the eligible articles. The inclusion criteria included: articles with full-text access, written in the English language, and focused on chronic sensory conditions and dietary flavonoid intake in an adult population. Studies focused on flavonoid supplements were excluded. Ten studies were included in this review. The evidence suggests that the flavonoid subclass, flavonols, are protective against eye conditions, including age-related macular degeneration, cataract, and glaucoma. There is insufficient evidence to support an association with hearing loss or tinnitus. Overall, dietary flavonol intake appears to be protective against some chronic eye conditions. However, for most eye and hearing-related conditions, only one study was identified. Thus, there is a need for more recent high-quality research to be conducted to confirm any significant associations.

## 1. Introduction

Flavonoids are bioactive polyphenolic compounds that are naturally ubiquitous in a range of plant-based produce, including fruits, vegetables, and beverages such as tea and red wine [1,2]. There are six major flavonoid subclasses, each with a number of key compounds—flavanols (e.g., epicatechin, epigallocatechin-gallate, and proanthocyanidins), flavanones (e.g., hesperidin and naringenin), flavones (e.g., apigenin and luteolin), isoflavones (e.g., genistein and daidzein), flavonols (e.g., quercetin, kaempferol, and myricetin) and anthocyanins (e.g., cyanidin and malvidin) [2,3]. Certain flavonoid subclasses are more abundant or limited to specific foods and beverages, for example, flavanones are common in citrus fruits; isoflavones in soy products; and anthocyanidins in wine and bilberry [1]. Due to the variability of flavonoids throughout the food supply, dietary intakes vary depending on food sources, food preferences, and dietary habits [4].

There has been growing research interest around the health benefits of flavonoids with evidence of antioxidant and anti-inflammatory properties as well as vascular health benefits through improved endothelial function [2,5]. This evidence has led to research around the potential benefits of flavonoids for vision and/or hearing loss and tinnitus, where their pathophysiology has been linked to oxidative stress, inflammation, and vascular health [6,7,8].

According to findings from the 2020 Global Burden of Disease (GBD) study [9], the leading causes of age-related vision loss in adults aged ≥50 years were cataract (15.2 million cases), glaucoma (3.6 million cases), age-related macular degeneration (AMD) (1.8 million cases), and diabetic retinopathy (0.86 million cases). To briefly describe each condition, cataract is a condition where the lens of the eye becomes progressively opaque, allowing less light to pass [10]; glaucoma is a condition characterised by changes to the optic nerve head and excavation of the optic disk causing the loss of the visual field and increased intraocular pressure [11,12,13]; AMD is a condition where the macula of the retina deteriorates leading to the loss of central vision [14]; and diabetic retinopathy is a complication of diabetes involving the presence of microaneurysms or haemorrhages in the retina [15].

The GBD study [16] also reported that 1.57 billion people in 2019 suffered from a hearing loss and that 62.1% of these people were aged ≥50 years. Among this older age group, ≥96.2% of hearing loss was age-related or caused by other factors such as noise exposure [16]. Age-related hearing loss (ARHL), also referred to as presbycusis, is a chronic condition involving the progressive loss of hearing [17]. People with ARHL often complain of tinnitus, a chronic auditory condition characterised by ringing or buzzing in the ears or head where an external source of sound is not present [18]. The prevalence of tinnitus steadily increases with age and ranges from 5.1 to 42.7% in adults [18].

Although treatments and therapies are available to address some of these chronic sensory conditions [9,14,16,19], the need to identify additional prevention and management strategies is warranted, not only because of the high prevalence rates, but also because vision and hearing loss are major causes of disability [9,16]. With reference to the GBD study, vision loss ranks as the eighth largest cause of disability in adults aged 50–69 years and the fourth largest cause in those aged 70+ years [9]. ARHL outranks vision loss in both older adult age groups as the third largest cause of disability [16].

With the promising health implications of flavonoids for vascular health and their research interest extending to chronic sensory conditions, this scoping review aims to summarise the current evidence in this area among adults. A scoping review was conducted to identify relevant studies and determine the gaps in research literature [20,21]. It also serves as a precursor to a systematic review by ensuring sufficient studies are available [21]. As such, scoping reviews do not exclude studies based on quality and, therefore, the inclusion and exclusion criteria are not strict. Using the included studies, the data are charted, and an optional consultation process with key stakeholders may follow.

## 2. Materials and Methods

A scoping review of literature was conducted based on Arksey and O’Malley’s six-stage framework [22]. This framework was the first methodological framework for conducting scoping reviews and has been widely used in the literature [22]. More recently, Levac and colleagues [23] refined this framework to ensure a consistent and systematic approach.

### 2.1. Search Strategy

MEDLINE, EMBASE PsycINFO (accessed via the OVID platform), and Google Scholar were the databases used to search for eligible studies to include in this review with no limits on the year of publication. The search strategy included the terms “Hearing loss OR hard of hearing OR deaf * OR hearing problems OR hearing health OR loss of hearing OR hearing impaired OR Hearing Disorder OR Tinnitus OR Cataracts OR glaucoma OR macular degeneration OR diabetic retinopathy OR dual sensory loss or dual sensory impairment AND flavonoids”. Reference lists of the identified articles were also manually searched.

### 2.2. Inclusion Criteria

Research articles of any study design written in the English language that investigated the associations between dietary flavonoids in relation to chronic eye conditions (cataracts, glaucoma, age-related macular degeneration, diabetic retinopathy) and/or hearing-related problems (hearing loss, tinnitus) in human adults were included in this review. Initially, the inclusion criteria also included supplemental flavonoids; however, this was a late exclusion following a 2021 publication of a systematic review reporting on the effects of flavonoid supplementation and chronic eye conditions to avoid repeating results [24]. There were no other exclusion criteria.

### 2.3. Screening Process

The screening criteria were established by the research team to reflect the inclusion criteria. They were refined after reviewing a sample of 20 abstracts and further refined prior to conducting full-text screening. The final screening criteria were: focuses on dietary flavonoids, chronic sensory conditions, an adult population, and presents empirical data. A pair of reviewers with a background in health sciences used the Rayyan tool [25] to independently screen the search results. Any conflicts were resolved by a third researcher (DT). The details of articles included for full-text screening were exported from Rayyan into a charting template. The reference list of included full-text articles were hand searched, along with the first five pages of the Google Scholar search results using the same search terms. The PRISMA flow chart outlines the screening process (Figure 1).

### 2.4. Data Extraction

The data extraction charting template was developed by research team members and captured details on study characteristics (e.g., title, author, year of publication, study design, and participants); outcomes and instruments; flavonoid details (i.e., type, form, and dose of flavonoids); and significant findings and recommendations. The same two reviewers with a health science background independently extracted the data from the included articles. A third researcher (DT) reviewed the completed charting template from each of the reviewers and resolved any conflicts by providing a majority judgement.

### 2.5. Quality Assessment

The Joanna Briggs Institute Critical Appraisal tools were used to assess the methodological rigor and bias of the included studies [26]. Two researchers (DT, YT) independently appraised each study and discussed any conflicts until both researchers agreed on the final outcome. These tools assess bias in a study’s design, conduct, and analysis [26]. The specific checklists used in this review included the checklist for analytical cross-sectional studies, checklist for case–control studies, and checklist for cohort studies [26]. Response options for each criterion in the checklists were ‘Yes’, ‘No’, ‘Unclear’, and ‘Not Applicable’. Conflicts between the researchers were discussed until the final quality assessment for each study was agreed upon.

## 3. Results

A total of 442 articles was initially identified from the OVID database and Google Scholar. After screening the title, abstract, and full-text article, and removing articles focused on supplemental flavonoids, 10 articles from six countries were identified. Specifically, this included Australia (n = 4), China (n = 1), Germany (n = 1), South Korea (n = 1), the Netherlands (n = 1), and the United States (n = 2). The quality assessment of the included studies showed that most of the respective criteria were satisfied (i.e., yes) (Appendix A). Ma et al.’s study [10] was the only case–control study and satisfied 8 out of 10 (80%) quality assessment criteria; one criterion (i.e., cases and controls matched appropriately) was not met and one criterion was unclear (i.e., was the exposure period long enough). For cross-sectional studies, Mahoney and Loprinzi’s study [15] satisfied all eight criteria (100%), while Kim et al.’s study [27] satisfied seven out of eight criteria (87.5%). The latter study did not satisfy the criterion about the exposure being measured in a valid and reliable way. Of the six cohort studies, studies by Ramdas et al. [13], Gopinath et al. [6], Kang et al. [12], Gopinath et al. [28], and Tang et al. [29] satisfied 9 out of 11 criteria (81.8%), while Detaram et al.’s study [30] satisfied 8 out of 11 criteria (72.7%). All six cohort studies either did not satisfy or were unclear about two criteria (i.e., indicating whether follow up was complete and reasons for loss to follow up; and indicating whether strategies to address incomplete follow up were utilized). In addition to these criteria, Detaram et al.’s study [30] did not meet the criteria about participants being free of the outcome. Terai et al.’s study [11] was the only randomised controlled trial and satisfied 12 out of 13 (92.3%) criteria. It did not satisfy the criteria about participants being blinded to treatment assignment.

### 3.1. Glaucoma

There were three studies reporting on the links between flavonoids and glaucoma: two on open-angle glaucoma [12,13] and one on unspecified glaucoma [11]. For open-angle glaucoma, both studies were prospective population-based cohort studies reporting on the usual dietary intake of flavonoids. Ramdas et al.’s Rotterdam study included 3502 participants, aged ≥55 years; an ophthalmic examination was performed to determine the presence of glaucoma and related measures (e.g., best-corrected visual acuity, visual field testing, and intraocular pressure); and dietary intake was assessed using a 170-item semiquantitative food frequency questionnaire that was administered by a trained dietitian [13]. Findings from this study suggested that dietary flavonoid intake was not significantly associated with the risk of open-angle glaucoma [13]. Kang et al.’s Nurses’ Health and Health Professionals Follow Up Study included 65,516 women and 42,156 men aged ≥40 years [12]. Similar to Ramdas et al.’s approach [13], glaucoma and related measures were assessed with ophthalmic examination and dietary intake assessed with a food frequency questionnaire; however, Kang et al.’s food frequency questionnaire was shorter, with 116+ items [12]. In Kang et al.’s study, a higher intakes of flavonols and monomeric flavanols marginally reduced the risk of open-angle glaucoma by 18% (RR = 0.82; 95% CI = 0.69, 0.97; *p*-trend = 0.05) and 14% (RR = 0.86; 95% CI = 0.72, 1.02; *p*-trend = 0.04), respectively [12]. The study also found that the consumption of approximately two cups of flavonoid-rich tea per day significantly reduced the risk of open-angle glaucoma by 18% ((RR = 0.82; 95% CI = 0.68, 0.99; *p*-trend = 0.02) [12].

Terai et al.’s study was the only one on unspecified glaucoma and investigated the impact of flavonoid-rich dark chocolate on retinal vessel diameter [11]. There were 30 glaucoma patients and 30 age-matched controls (aged ≥50 years). As with the other two studies, glaucoma and related measures were assessed through an ophthalmic examination. Participants were randomly assigned to either dark or white chocolate and provided standardised doses after completing a cacao-free period for 24 h and a 12 h overnight fast [11]. This randomised controlled trial reported a significant increase in venous vasodilation (3.2 ± 0.9% to 4.2 ± 1.4%; *p* = 0.01) after dark chocolate consumption among control participants. No significant changes were observed among glaucoma cases.

### 3.2. Diabetic Retinopathy

One study by Mahoney and Loprinzi investigated the associations between diabetic retinopathy and flavonoids through fruit and vegetable consumption [15]. This study included 381 diabetic participants from the National Health and Nutrition Examination Survey (NHANES), 2003–2006. The presence of diabetic retinopathy was determined with a retinal examination and evaluated against the Early Treatment Diabetic Retinopathy Study (ETDRS) grading criteria [15]. Dietary intake in the last 12 months was assessed using a food frequency questionnaire [15]. Mahoney and Loprinzi reported that higher intakes of flavonoid-rich fruits and vegetables were associated with significantly lower C-reactive protein (β = −0.005), HbA1C (β = −0.008), and glucose levels (β = −0.51), and 33% reduced odds of having diabetic retinopathy (OR = 0.67, 95% CI: 0.45–0.99; *p* = 0.04) [15].

### 3.3. Age-Related Macular Degeneration (AMD)

Three studies explored the association between dietary flavonoids and AMD. Of these, there were two population-based studies [6,27], including one from our group [6]. Our study, involving 2856 adults aged ≥49 years at baseline, assessed usual intake using a semiquantitative food frequency questionnaire and determined the incidence of AMD from retinal photographs over a 15-year period [6]. Reported findings included a reduced risk of any AMD with each one standard deviation (SD) increase total flavonoid intake (OR = 0.76; 95% CI: 0.58, 0.99); reduced odds of prevalent AMD with each 1-SD increase in flavonol and flavanone intakes (OR = 0.75; 95% CI: 0.58, 0.97) and OR = 0.77; 95% CI: 0.60, 0.99, respectively); and a reduced risk of late AMD after 15 years among participants who consumed ≥1 serving of oranges/day compared to those who never consumed oranges at baseline (OR = 0.39; 95% CI: 0.18, 0.85) [6]. The second population-based study was conducted by Kim et al. in South Korea [27]. There were 1008 women aged ≥65 years, AMD was assessed using fundus photographs, and dietary intake was collected through face-to-face 24 h recalls. As with our study [6], Kim and colleagues also reported significant associations between dietary flavonol intake and prevalent AMD, reporting reduced odds of 55% (OR: 0.45; 95% CI: 0.25–0.82, *p*-trend = 0.008) [27].

The third study investigating dietary flavonoid intake and treatment outcomes in neovascular AMD patients was a clinic-based cohort study conducted by our group [30]. In 547 AMD patients at baseline, AMD outcomes were assessed through ophthalmic examination and dietary intake was assessed using the same food frequency questionnaire as our previous study [6]. We found that participants with lower intakes of flavonol (specifically quercetin), flavan-3-ols (specifically epigallocatechin-3-gallate and epigallocatechin), and flavonoid-rich tea were associated with significantly poorer vision (multivariable-adjusted least-square mean visual acuity; 14.68 vs. 19.53 (*p* = 0.04); 14.06 vs. 18.89 (*p* = 0.04); 13.86 vs. 18.86 (*p* = 0.03); and 13.49 vs. 19.04, *p* = 0.02, respectively) [30]. Our study also found that lower intakes of flavan-3-ols, epigallocatechin-3-gallate, and epigallocatechin were associated with twice the risk of having intraretinal fluid (multivariable-adjusted *p* trend of 0.03, 0.01 0.02, respectively) [30]. As tea is a rich source of dietary flavonoids, Deteram et al. [30] conducted additional analyses to show that the lowest versus highest tertile of tea consumption increased the odds of intraretinal fluid presence by 2.13 times (OR = 2.13, CI: 1.18–3.85).

### 3.4. Cataract

One study investigated the links between flavonoid intake and risk of age-related cataract [10]. This population-based cohort study included adults aged 50 to 70 years (n = 249 cases, 66 controls) with the presence of cataract determined by slit-lamp examination. Dietary intake was assessed using a semiquantitative food frequency questionnaire. A subsample of participants also participated in an embedded validity study that involved completing three 24 h dietary records to compare against responses to the food frequency questionnaire. Ma et al. [10] reported that lower versus higher intakes of flavonols (quercetin and isorhamnetin) increased cataract risk by over 11-fold (OR 11.78, 95% CI: 1.62–85.84, *p* < 0.05) and almost 7-fold (OR 6.99, 95% CI: 1.12–43.44, *p* < 0.05), respectively. However, the authors noted that isorhamnetin is sourced from a limited number of foods and this finding may have been due to chance [10].

### 3.5. Hearing Loss

A study by our group was the only study found to investigate associations between dietary flavonoids and hearing loss. This population-based cohort study included 1691 participants at baseline. Pure-tone audiometry was assessed in soundproof booths at 0.5 to 4 KHz and a measure of >25 dB hearing level indicated the presence of hearing loss. Usual dietary intake in the last 12 months was captured with a 145-item semiquantitative food frequency questionnaire. It was observed that there was a 36% lower risk of incident hearing loss after 10 years with higher intakes of isoflavones (OR = 0.64; 95% CI, 0.42–0.99; *p*-trend = 0.03) [28]. However, Gopinath et al. [28] noted that isoflavones are primarily sourced from soy-based products, which were not widely consumed by the study population, and, therefore, acknowledged that this may be a chance finding. No other significant associations were observed. It was noted that the limitations of this study included the use of an American flavonoids database which may not have accurately captured the content in Australian food, and that other confounding variables that were not assessed in this study may have influenced the results [28].

### 3.6. Tinnitus

Similarly, a study by our group was the only study found to investigate associations between dietary flavonoids and tinnitus [29]. Our population-based cohort study included 1753 participants aged 50+ years, where the presence of tinnitus was determined by an audiologist-administered question asking participants “Have you experienced any prolonged ringing, buzzing, or other sounds in your ears or head within the past year that is lasting for five minutes or longer?” and the usual dietary intake in the last 12 months was captured by a 145-item semiquantitative food frequency questionnaire. It was concluded that there was insufficient evidence to support a protective association between the development of incident tinnitus and dietary flavonoid intake [29]. Similar limitations as our study on hearing loss [28] were reported.

## 4. Discussion

To the best of our knowledge, this is the first scoping review to summarise the current evidence on the associations between dietary flavonoid intake and chronic sensory conditions in adults. Six out of eight studies reported favourable associations between dietary flavonoid intake with chronic eye conditions [6,10,12,15,27,30]. In particular, higher intakes of dietary flavonols were associated with a reduced risk of primary open-angle glaucoma [12] and reduced odds of prevalent AMD [6,30], while lower intakes of dietary flavonols were associated with poorer visual acuity related to AMD [30] and increased risk of age-related cataract [10]. For glaucoma, the role of flavonols in risk reduction may be linked to antioxidant activity. This is because the pathogenesis of glaucoma and its common symptom (i.e., higher intraocular pressure) have been associated with oxidative damage to the trabecular meshwork and surrounding endothelial cells [31]. In an experimental investigation by Miyamoto et al. [31], the flavonol quercetin was shown to induce the expression of antioxidant enzymes (peroxiredoxins) to reduce oxidative damage and thereby regulate intraocular pressure levels to normal levels [12,31]. For AMD, flavonols have been linked to a number of pathways. One example includes the regulation of the nitric oxide status by enhancing its production and increasing the amount of circulating nitrite in order to improve endothelial function [5,6,30]. A second example of the potential role of flavonols in AMD includes scavenging reactive oxygen species that may be causing oxidative damage to the retinal pigment epithelium cells [6,30,32]. The integrity of the retinal pigment epithelium is important to the health of the macula and, thus, in the prevention of AMD development and/or progression [14]. The third example includes the inhibition of retinal and choroidal angiogenesis [6,30,33,34]. Experimental studies conducted in vitro have shown that quercetin inhibits angiogenesis by impacting the development and spread of new blood vessels within the choroid and retina regions of the eye [33,34]. For cataract development, flavonols are suggested to play a role in modifying pathways causing eye lens opacification, including oxidative stress, epithelial function, nonenzymatic glycation, the polyol pathway, and lens calpain proteases [10,35]. The antioxidant role of flavonoids such as flavonols has been linked to protecting and promoting antioxidant enzymes and inhibiting the function of the enzyme aldose reductase, which has a key role in the development of cataract among patients with diabetes [36].

In comparison to Davinelli’s recent systematic review and meta-analysis on flavonoid supplementation and common eye conditions, a total of 11 studies was included, of which there were 4 studies on diabetic retinopathy, 6 studies on glaucoma and intraocular pressures, and 1 study on dry eye disease [24]. In contrast to our review on dietary flavonoids, there were no eligible studies investigating the effects of flavonoid supplementation on age-related macular degeneration and age-related cataract. A second contrasting finding between our reviews was that Davinelli et al.’s findings did not show that flavonols were beneficial for eye health. Instead, supplemental flavan-3-ols (standard mean difference = −0.62; 95% CI: −1.03, −0.22, *p* < 0.01) and anthocyanins (standard mean difference = −0.42; 95% CI: −0.63, −0.21, *p* < 0.01) were the only flavonoid subclasses to demonstrate significant improvement in eye health [24]. These improvements were observed with glaucoma and intraocular pressure, and diabetic retinopathy [24]. In our review, one study reported a significant benefit of dietary flavan-3-ols in relation to AMD outcomes [30]. According to Detaram et al. [30], lower intakes of dietary flavan-3-ols were significantly associated with poorer vision and almost double the risk of intraretinal fluid compared with higher intakes of flavon-3-ols. We suspect that the lack of supporting evidence around the protective effects of dietary flavan-3-ols and anthocyanins for eye health in our review may be due to their lower availability in the diet in comparison to flavonols [37,38], particularly the flavonol quercetin, which has been reported to be the most widely consumed and studied flavonoid [10].

Contrastingly, the evidence from this review does not support an association between dietary flavonoid intake and hearing loss or tinnitus in adults. However, it is worth mentioning that only one study for each condition was identified in this review, and that both of these studies reported on data from the Blue Mountains Hearing Study [28,29]. Although the Blue Mountains Hearing Study is a large cohort study, it is based on the Australian population and findings may not be generalisable to other countries. Moreover, the last period of data collection for the Blue Mountains Hearing Study was in 2007–2009 [28,29], and so more recent studies are needed to capture any changes within the population relating to factors such as demographics, eating patterns, as well as technological advancement in the assessment of outcomes related to hearing loss and tinnitus. Protective associations between dietary flavonoids and hearing loss and tinnitus are plausible, as oxidative stress and inflammatory pathways have been linked to causing cochlear damage [39], and the role of flavonoids in the improvement of endothelial function could influence the integrity of the capillary bed within the cochlea [5].

### 4.1. Recommendations

This scoping review highlighted the lack of research investigating the associations between dietary flavonoids and chronic sensory conditions. As most conditions were limited to evidence from one study, with the exception of glaucoma and AMD, where three studies were identified for each, further research in this area is warranted. A potential starting point includes conducting additional high-quality studies to confirm or disprove the promising findings related to dietary flavonol intake and eye health identified in this review. Specifically, there is a need for evidence from randomised controlled trials, as only one study of this design was identified in this review. With more research in this area, there may be a sufficient number of studies to conduct a systematic review and meta-analyses in the future to provide more conclusive evidence on the association between dietary flavonoids and sensory conditions.

### 4.2. Limitations

Limitations of this review include the small number of eligible studies identified. However, as the purpose of this scoping review was to identify the existing number of studies and the gaps in research literature on this topic, the few studies yielded emphasises the clear lack of evidence and importance of further research. As such, this statement echoes our primary recommendation. We also acknowledge that all relevant studies may not have been identified by our review method. For example, studies published in languages other than English, and studies published in subscription-based journals that our institutions do not have access to would have been excluded. Further, the optional sixth stage of Arksey and O’Malley’s scoping review framework [22] (i.e., consultation with key stakeholders) was not included, as we were unable to sample a wide range of opinions from key stakeholders to ensure an unbiased perspective. However, we hope clinicians and researchers engage with this article and further explore the potential links between dietary flavonoids and sensory conditions.

## 5. Conclusions

Overall, this review identified evidence supporting a protective association between dietary flavonoids, particularly flavonols, and AMD, glaucoma, and cataract. However, for most of the sensory conditions included in this review, only one study was identified. This signifies a strong need for further research in this area to substantiate any associations.

## Figures and Tables

**Figure 1 antioxidants-11-01214-f001:**
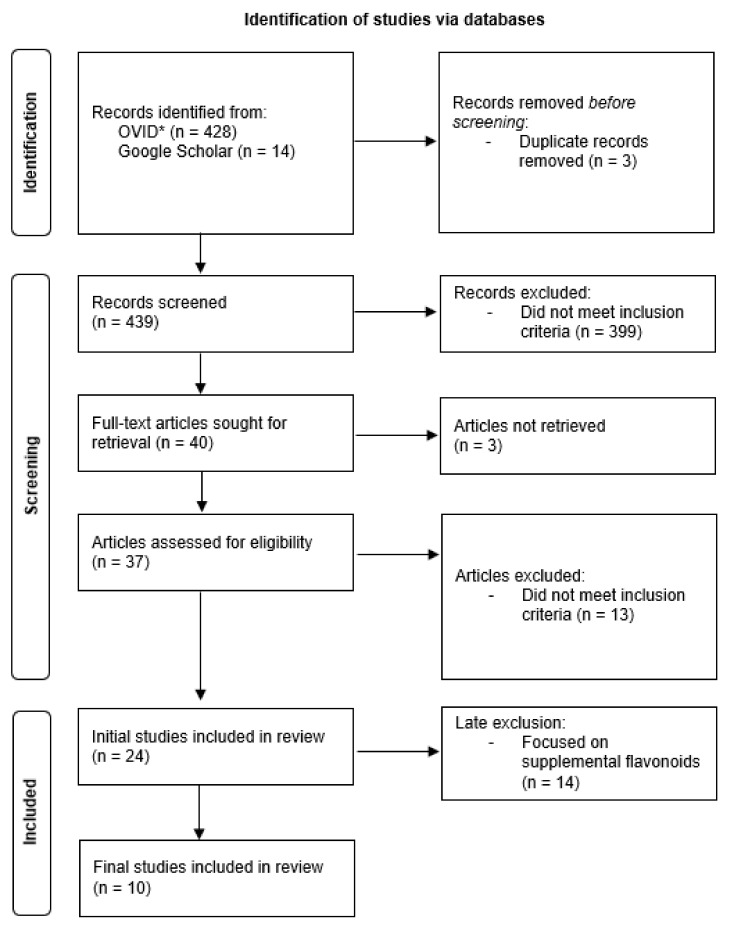
PRISMA flowchart of the literature screening process. * Includes MEDLINE, EMBASE, and PsycINFO databases.

## Data Availability

The data are contained within the article and Appendix A.

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
