# Peer review of "Dietary Flavonoid Intake and Chronic Sensory Conditions: A Scoping Review"

_antioxidants, 2022, doi:10.3390/antiox11071214_

Round 1

Reviewer 1 Report

In this manuscript, Tang et al. aim to review the protective potential of the dietary intake of flavonoids in hearing and eye-related diseases. The topics fall within the scope of the journal and the study of the effects of dietary flavonoid intake could be of interest to the journal’s readers. However, the major limitation of the manuscript is the very few eligible papers to perform the analysis. The author should explain if the number is enough to perform a scoping review. In other words, does it make sense to perform a scoping review of a topic that is featured by a lack of literature?

The manuscript is well-written and follows an organized structure. The manuscript is easy to read because the writing is clear. Some comments should be addressed:

• The first time the authors introduce the Arksey and O’Malley six-stage framework (line 14), they should explain briefly what is the significance to use this method for the scoping review (in comparison to other available methods).

• It is highly desirable a Table that compares the conclusions of the potential of flavonoid supplementation (ref18) and the conclusion drawn in this scoping review.

• Tables 1 and 2 are unnecessary. They should be deleted.

• There is no legend in Figure1.

• In line 13, the authors should write “align” and not “aligns”

• When the authors introduce a number in lines 52, 53, and 54, they write it as 3·6 million. They should rewrite it since it is not clear what they mean (3.6 million?. 3-6 million?)

• In lines 60-61, do the authors refer to “an external source of sound”?

• What do the authors mean to say in line 100 “the screening criteria were established by the research team”? How were the criteria established? Which ones are the exclusion criteria (line 131)? These aspects should be clearly defined.

• What do “DT” and “YT” mean? line 103; line 118; line 121.

• The authors should define SD (line 168)

• The references should be revised because some mistakes were found and they should fulfill the format to be published in the journal. In addition, some refs should be replaced with other ones more updated and relevant

Author Response

Reviewer 1

Comment 1. However, the major limitation of the manuscript is the very few eligible papers to perform the analysis. The author should explain if the number is enough to perform a scoping review. In other words, does it make sense to perform a scoping review of a topic that is featured by a lack of literature?

Response 1. We would like to thank Reviewer 1 for taking the time to review our manuscript. We acknowledge that only 10 eligible papers were identified in this scoping review, however, several scoping reviews have previously been conducted with similar or fewer papers. We have included the references of some examples below. Moreover, according to Munn et al. 2018’s publication ‘Systematic review or scoping review? Guidance for authors when choosing between a systematic or scoping review approach’ scoping reviews have a number of purposes, which include being conducted as precursors to systematic reviews to ensure sufficient studies are available. Therefore, the small number of studies identified in this scoping review is a useful indicator to avoid conducting a systematic review on this topic at present. We have revised our description of scoping reviews in the introduction section as below.

Line 77: “A scoping review was conducted to identify relevant studies and determine the gaps in research literature [20,21]. It also serves as a precursor to a systematic review by ensuring sufficient studies are available [21].”

Examples of other scoping reviews:

  • Anggi Lukman Wicaksana, Nuzul Sri Hertanti et al. Diabetes management and specific considerations for patients with diabetes during coronavirus diseases pandemic: A scoping review. Diabetes & Metabolic Syndrome: Clinical Research & Reviews, 2020-09-01, Volume 14, Issue 5, Pages 1109-1120. N=7 papers
  • Huard Pelletier V, Lessard A, Piché F, et al. Video games and their associations with physical health: a scoping reviewBMJ Open Sport & Exercise Medicine 2020;6:e000832. N=12 papers
  • Åsberg, K., Bendtsen, M. Perioperative digital behaviour change interventions for reducing alcohol consumption, improving dietary intake, increasing physical activity and smoking cessation: a scoping review. Perioper Med 10, 18 (2021). N=11 papers
  • Muhammad H, Reeves S, Ishaq S, Jeanes Y. Interventions to Increase Adherence to a Gluten Free Diet in Patients with Coeliac Disease: A Scoping Review. Gastrointestinal Disorders. 2020; 2(3):318-326. N=11 papers
  • Vaiciurgis, V., Charlton, K., Clancy, A., & Beck, E. (2022). Nutrition programmes for individuals living with disadvantage in supported residential settings: A scoping review. Public Health Nutrition, 1-12. N=7 papers

Comment 2. The manuscript is well-written and follows an organized structure. The manuscript is easy to read because the writing is clear. Some comments should be addressed: The first time the authors introduce the Arksey and O’Malley six-stage framework (line 14), they should explain briefly what is the significance to use this method for the scoping review (in comparison to other available methods).

 Response 2. Thank you for the suggestion to explain the Arksey and O’Malley framework. This has now been added as below.

Line 84: “A scoping review of literature was conducted based on Arksey and O’Malley’s six stage framework [22]. This framework was the first methodological framework for con-ducting scoping reviews and has been widely used in the literature [22]. More recently, Levac and colleagues [23] have refined this framework to ensure a consistent and systematic approach.”

Comment 3. It is highly desirable a Table that compares the conclusions of the potential of flavonoid supplementation (ref18) and the conclusion drawn in this scoping review.

Response 3. Thank you for the suggestion to include a comparison table. Davinelli and colleagues only found studies on diabetic retinopathy, glaucoma and dry eye disease in relation to flavonoid supplementation. As only diabetic retinopathy and glaucoma overlap with the conditions in our study, we felt that a more thorough comparison within the discussion section of this manuscript was more appropriate.

Line 309 “In comparison to Davinelli’s recent systematic review and meta-analysis on fla-vonoid supplementation and common eye conditions, a total of 11 studies were included, of which there were 4 studies on diabetic retinopathy, 6 studies on glaucoma and intraocular pressures and 1 study on dry eye disease [24]. In contrast to our review on dietary flavonoids, there were no eligible studies investigating the effects of flavonoid supplementation on age-related macular degeneration and age-related cataract. A second contrasting finding between our reviews, is that Davinelli et al.’s findings did not show that flavonols were beneficial for eye health. Instead, supplemental flavan-3-ols (standard mean difference = −0.62; 95% CI: −1.03, −0.22, p < 0.01) and anthocyanins (standard mean difference = −0.42; 95% CI: −0.63, −0.21, p < 0.01) were the only flavonoid subclasses to demonstrate significant improvement in eye health [24]. These improvements were observed with glaucoma and intraocular pressure, and diabetic retinopathy [24]. In our review, one study reported a significant benefit of dietary flavan-3-ols in relation to AMD outcomes [30]. According to Detaram et al [30], lower intakes of dietary flavan-3-ols were significantly associated with poorer vision and almost double the risk of intraretinal fluid compared with higher intakes of flavan-3-ols. We suspect that the lack of supporting evidence around the protective effects of dietary flavan-3-ols and anthocyanins for eye health in our review may be due to their lower availability in the diet in comparison to of flavonols [37,38], particularly the flavonol - quercetin, which has been reported to be the most widely consumed and studied flavonoid [10].”

Comment 4. Tables 1 and 2 are unnecessary. They should be deleted.

Response 4. Thank you for your suggestion. We have deleted Table 1 and 2.

Comment 5. There is no legend in Figure1.

Response 5. Thank you for identifying this discrepancy. A legend has now been included stating “Fig 1. PRISMA flowchart of the literature screening process.”

Comment 6. In line 13, the authors should write “align” and not “aligns”

Response 6. Thank you for identifying this grammatical error. This has now been corrected.

Comment 7. When the authors introduce a number in lines 52, 53, and 54, they write it as 3·6 million. They should rewrite it since it is not clear what they mean (3.6 million?. 3-6 million?)

Response 7. Thank you for identifying this formatting error. This has now been corrected (e.g., 3.6 million cases).

Comment 8. In lines 60-61, do the authors refer to “an external source of sound”?

Response 8. Apologies for the lack of clarity. Yes, we refer to an external source of sound and this has now been specified.

Comment 9a. What do the authors mean to say in line 100 “the screening criteria were established by the research team”? How were the criteria established?

Response 9. Apologies for the lack of explanation regarding the screening criteria. This has now been included as below.

Line 110: “The screening criteria were established by the research team to reflect the inclusion criteria. They were refined after reviewing a sample of 20 abstracts and further refined prior to conducting full text screening. The final screening criteria were: focuses on dietary flavonoids, chronic sensory conditions, and an adult population, and presents empirical data.”

Comment 9b. Which ones are the exclusion criteria (line 131)? These aspects should be clearly defined.

Response 9b. Apologies for the lack of clarity regarding the exclusion criteria. As mentioned in the inclusion criteria paragraph, supplemental flavonoids were a late exclusion following the publication of the systematic review on supplemental flavonoids and common eye conditions. No other exclusion criteria were specified. The inclusion criteria paragraph has been reworded to indicate the final inclusion criteria, as below.

Line 100: “Research articles of any study design, written in the English language that investigated the associations between dietary flavonoids in relation to chronic eye conditions (cataracts, glaucoma, age-related macular degeneration, diabetic retinopathy) and/or hearing-related problems (hearing loss, tinnitus) in human adults were included in this review. Initially, the inclusion criteria also included supplemental flavonoids, however, this was a late exclusion following a 2021 publication of a systematic review reporting on the effects of flavonoid supplementation and chronic eye conditions to avoid repeating results [24]. There were no other exclusion criteria.”

Comment 10. What do “DT” and “YT” mean? line 103; line 118; line 121.

Response 10. Thank you for your comment. DT and YT are abbreviations of the authors’ names (i.e., Diana Tang and Yvonne Tran). This is used to indicate which author was responsible for the mentioned tasks.

Comment 11. The authors should define SD (line 168)

Response 11. Thank you for your suggestion. We have now defined SD as “standard deviation”.

Comment 12. The references should be revised because some mistakes were found and they should fulfill the format to be published in the journal. In addition, some refs should be replaced with other ones more updated and relevant

Response 12. We would like to thank Reviewer 1 for taking the time to thoroughly review this manuscript and provide constructive feedback.  We have taken on board your suggestion to review and revise the reference list to align with the journal’s format and include more recent references.

Reviewer 2 Report

The authors have conducted a scoping review on the effect of dietary flavonoid intake on ocular and hearing pathologies. Among the conclusions, they find an effect of the flavonols subclass on ocular protection, including age-related macular degeneration, cataract, and glaucoma. However, they do not see sufficient evidence to support an association with hearing damage.

I think it is an interesting article, however, they have had to eliminate a large number of articles related to flavonoid supplementation and diseases which, due to the existence of a systematic review in 2021, so the number of final selected articles is not very high.

There are aspects that detract from the scientific validity of this review, such as the low final number of articles together with the diversity of the type of studies present (case-control, cohort, etc.) or the mixing of studies in which a flavonoid-rich food is given with studies based on data from different types of nutritional surveys. In addition, the authors deal with a wide variety of ocular pathologies (Glaucoma, Diabetic retinopathy, Age-related macular degeneration (AMD), Cataract), it results in conclusions being drawn from only one study.

In addition, a percentage of plagiarism higher than 35% has been detected, although this is fundamentally in the tables in quality, it should be reviewed, if possible.

But despite these drawbacks, the article is still interesting, focusing only on the effect of a dietary intake of flavonoids and showing the absence of these studies and the need to go deeper into these possible effects.

One aspect that should be improved is the description of the final inclusion criteria; it is indicated that the concept of supplementation is eliminated. Also, the screening criteria selected by the research team.

Also, since this is a journal with a clear focus on oxidative aggression, I think it would be interesting to give some information on the mechanisms through which these flavonoids may act on the ocular and auditory diseases studied, focusing on the antioxidative field.

Author Response

Comment 1. I think it is an interesting article, however, they have had to eliminate a large number of articles related to flavonoid supplementation and diseases which, due to the existence of a systematic review in 2021, so the number of final selected articles is not very high.

Response 1. We would like to thank Reviewer 2 for taking the time to review our manuscript. We acknowledge that only 10 eligible papers were identified in this scoping review, however, several scoping reviews have previously been conducted with similar or fewer papers. We have included the references of some examples below. Moreover, according to Munn et al. 2018’s publication ‘Systematic review or scoping review? Guidance for authors when choosing between a systematic or scoping review approach’ scoping reviews have a number of purposes, which include being conducted as precursors to systematic reviews to ensure sufficient studies are available. Therefore, the small number of studies identified in this scoping review is a useful indicator to avoid conducting a systematic review on this topic at present. We have revised our description of scoping reviews in the introduction section as below.

Line 77: “A scoping review was conducted to identify relevant studies and determine the gaps in research literature [20,21]. It also serves as a precursor to a systematic review by ensuring sufficient studies are available [21].”

Examples of other scoping reviews:

  • Anggi Lukman Wicaksana, Nuzul Sri Hertanti et al. Diabetes management and specific considerations for patients with diabetes during coronavirus diseases pandemic: A scoping review. Diabetes & Metabolic Syndrome: Clinical Research & Reviews, 2020-09-01, Volume 14, Issue 5, Pages 1109-1120 – n=7 papers
  • Huard Pelletier V, Lessard A, Piché F, et al. Video games and their associations with physical health: a scoping reviewBMJ Open Sport & Exercise Medicine 2020;6:e000832. Doi: 10.1136/bmjsem-2020-000832 – n=12 papers
  • Åsberg, K., Bendtsen, M. Perioperative digital behaviour change interventions for reducing alcohol consumption, improving dietary intake, increasing physical activity and smoking cessation: a scoping review. Perioper Med 10, 18 (2021). https://doi.org/10.1186/s13741-021-00189-1 - n=11 papers
  • Muhammad H, Reeves S, Ishaq S, Jeanes Y. Interventions to Increase Adherence to a Gluten Free Diet in Patients with Coeliac Disease: A Scoping Review. Gastrointestinal Disorders. 2020; 2(3):318-326. https://doi.org/10.3390/gidisord2030029 - n=11 papers
  • Vaiciurgis, V., Charlton, K., Clancy, A., & Beck, E. (2022). Nutrition programmes for individuals living with disadvantage in supported residential settings: A scoping review. Public Health Nutrition, 1-12. Doi:10.1017/S1368980022000969 – n=7 papers

Comment 2. There are aspects that detract from the scientific validity of this review, such as the low final number of articles together with the diversity of the type of studies present (case-control, cohort, etc.) or the mixing of studies in which a flavonoid-rich food is given with studies based on data from different types of nutritional surveys. In addition, the authors deal with a wide variety of ocular pathologies (Glaucoma, Diabetic retinopathy, Age-related macular degeneration (AMD), Cataract), it results in conclusions being drawn from only one study.

Response 2. Thank you for your comment. As addressed above in Response 1, the purpose of this scoping review is to provide an overview of existing studies on this topic to identify current gaps in knowledge and inform the suitability of the current literature to conduct future systematic reviews and meta-analyses. Therefore, we hope that with the small number of identified studies on dietary intakes of flavonoids and a range of chronic sensory conditions, it provides useful insight to the readers that there is insufficient literature to proceed with systematic reviews focused on one specific condition or a specific method of dietary assessment, at this stage. We have now added this to the manuscript under the recommendations and limitations sections, respectively, as below.

Line 351: “With more research in this area, there may be a sufficient number of studies to conduct a systematic review and meta-analyses in the future to provide more conclusive evidence on the association between dietary flavonoids and sensory conditions.”

Line 356: “Limitations of this review include the small number of eligible studies identified. However, as the purpose of this scoping review is to identify the existing number of studies and the gaps in research literature on this topic, the few studies yielded emphasises the clear lack of evidence and importance of further research. As such, this statement echoes our primary recommendation.”

Comment 3. In addition, a percentage of plagiarism higher than 35% has been detected, although this is fundamentally in the tables in quality, it should be reviewed, if possible.

Response 3. Thank you for your suggestion. We have reviewed the manuscript in an attempt to reduce the percentage of plagiarism. We note that the contents of the tables cannot be reworded as they are the quality assessment criteria of the tools used and these results have been presented in a similar way to a recent publication by Antioxidants (https://www.mdpi.com/2076-3921/10/11/1841/htm). Therefore, we have now moved these tables to Supplementary File 1 and provided a worded summary in its place. We have now also included a reference to the Joanna Briggs Institute Critical Appraisal Tools for each of the quality tables.

Comment 4a. One aspect that should be improved is the description of the final inclusion criteria; it is indicated that the concept of supplementation is eliminated.

Response 4a. Thank you for your suggestion. The inclusion criteria have been reworded to clarify the final inclusion criteria, as below.

Line 100: “Research articles of any study design, written in the English language that investigated the associations between dietary flavonoids in relation to chronic eye conditions (cataracts, glaucoma, age-related macular degeneration, diabetic retinopathy) and/or hearing-related problems (hearing loss, tinnitus) in human adults were included in this review. Initially, the inclusion criteria also included supplemental flavonoids, however this was a late exclusion following a 2021 publication of a systematic review reporting on the effects of flavonoid supplementation and chronic eye conditions to avoid repeating results [24]. There were no other exclusion criteria.”

Comment 4b. Also, the screening criteria selected by the research team.

Response 4b. Apologies for the lack of explanation regarding the screening criteria. This has now been included as below.

Line 110: “The screening criteria were established by the research team to reflect the inclusion criteria. They were refined after reviewing a sample of 20 abstracts and further refined prior to conducting full text screening. The final screening criteria were: focuses on dietary flavonoids, chronic sensory conditions, and an adult population, and presents empirical data.”

Comment 5. Also, since this is a journal with a clear focus on oxidative aggression, I think it would be interesting to give some information on the mechanisms through which these flavonoids may act on the ocular and auditory diseases studied, focusing on the antioxidative field.

Response 5. Thank you for all your feedback to improve this manuscript for publication. We have taken on board your suggestion and have now elaborated on the potential antioxidative mechanism of flavonoids in eye health in the discussion section of this manuscript as below.

Line 286: “For glaucoma, the role of flavonols in risk reduction may be linked to antioxidant activity. This is because the pathogenesis of glaucoma and its common symptom (i.e., higher intraocular pressure) have been associated with oxidative damage to the trabecular meshwork and surrounding endothelial cells [31]. In an experimental investigation by Miyamoto et al. [31], the flavonol - quercetin, has been shown to induce the expression of antioxidant enzymes (peroxiredoxins) to reduce oxidative damage and thereby regulate intraocular pressure levels to normal levels [12,31].

Line 292: “For AMD, flavonols have been linked to a number of pathways. One example includes regulation of nitric oxide status by enhancing its production and increasing the amount of circulating nitrite in order to improve endothelial function [5,6,30]. A second example of the potential role of flavonols in AMD include scavenging reactive oxygen species that may be causing oxidative damage to the retinal pigment epithelium cells [6,30,32]. The integrity of the retinal pigment epithelium is important to the health of the macula and thus in the prevention of AMD development and/or progression [14]. The third example includes inhibition of retinal and choroidal angiogenesis [6,30,33,34]. Experimental studies conducted in-vitro have shown that quercetin inhibits angiogenesis by impacting the development and spread of new blood vessels within the choroid and retina regions of the eye [33,34].”

Line 302: “For cataract development, flavonols are suggested to play a role in modifying pathways causing eye lens opacification including oxidative stress, epithelial function, non-enzymatic glycation, the polyol pathway and lens calpain proteases [10,35]. The antioxidant role of flavonoids like flavonols, has been linked to protecting and promoting antioxidant enzymes, and inhibiting the function of enzyme, aldose reductase, which has a key role in the development of cataract among patients with diabetes [36].”